# Unleashing the Potential of Medicinal Plants in Benin: Assessing the Status of Research and the Need for Enhanced Practices

**DOI:** 10.3390/plants12071506

**Published:** 2023-03-30

**Authors:** Eric Agbodjento, Boris Lègba, Victorien Tamègnon Dougnon, Jean Robert Klotoé, Esther Déguénon, Phénix Assogba, Hornel Koudokpon, Leena Hanski, Lamine Baba-Moussa, Eléonore Yayi Ladékan

**Affiliations:** 1Research Unit in Applied Microbiology and Pharmacology of Natural Substances, Research Laboratory in Applied Biology, Polytechnic School of Abomey-Calavi, University of Abomey-Calavi, Cotonou 01 PB 2009, Benin; 2Drug Research Program, Division of Pharmaceutical Biosciences, Faculty of Pharmacy, University of Helsinki, 00100 Helsinki, Finland; 3Laboratory of Biology and Molecular Typing in Microbiology, Department of Biochemistry and Cellular Biology, Faculty of Sciences and Techniques, University of Abomey-Calavi, Cotonou 05 PB 1604, Benin; 4Laboratory of Pharmacognosy and Essential Oils, Faculty of Sciences and Techniques, University of Abomey-Calavi, Cotonou 01 PB 188, Benin

**Keywords:** phytomedicine, scientific investigations, state of the art, pharmacology, toxicology, phytochemistry

## Abstract

Medicinal plants play a crucial role in the primary health care of the population in developing countries such as Benin. The national universities of Benin conduct research on the pharmacological, toxicological, and phytochemical properties of these plants, but the resulting knowledge often does not lead to the development of phytomedicines for the improvement of public health. This study aims to assess the current status of research on medicinal plants in Benin. A literature review was conducted using various search engines, and the collected data was analyzed. The results indicate that research on the biological properties of medicinal plants is still in its early stages, with limited and inadequate methodological approaches. These limitations highlight the urgent need to enhance research practices and facilitate the development of effective and safe phytomedicines.

## 1. Introduction

In many developing countries, including Benin, traditional medicine plays a crucial role in providing primary health care to the population. The abundant use of medicinal plants in traditional medicine and the transmission of this knowledge from generation to generation highlights the importance of preserving and utilizing this resource in a sustainable manner.

Benin is a country in West Africa with a rich tradition of herbal medicine. Many plants are used by the population of Benin for various purposes, including medicinal, culinary, and spiritual uses. Here are some of the most commonly used plants in Benin [1]:-*Aloe vera*: *Aloe vera* is a succulent plant that is commonly used for medicinal purposes. In Benin, it is used to treat various ailments, including skin conditions, digestive problems, and infections.-*Neem*: *Neem* is a tree that is widely used in Benin for its medicinal properties. It is used to treat a variety of conditions, including fever, malaria, and skin diseases.-*Hibiscus*: *Hibiscus* is a plant that is used in Benin to make a popular drink known as bissap. The drink is made by steeping the leaves of the hibiscus plant in hot water and adding sugar or honey. It is believed to have a range of health benefits, including reducing high blood pressure and boosting the immune system.-*Moringa*: *Moringa* is a tree that is used in Benin for its nutritional and medicinal properties. It is rich in vitamins and minerals and is used to treat a variety of conditions, including anemia, diabetes, and high blood pressure.-*Baobab*: *Baobab* is a tree that is native to Africa and is used in Benin for its medicinal and nutritional properties. The fruit of the baobab tree is rich in vitamin C, fiber, and antioxidants, and is used to treat a variety of conditions, including diarrhea, constipation, and respiratory infections.-*Ginger*: *Ginger* is a root that is widely used in Benin for its medicinal and culinary properties. It is used to treat a variety of conditions, including nausea, indigestion, and inflammation.-*Turmeric*: *Turmeric* is a spice that is widely used in Benin for its medicinal and culinary properties. It is used to treat a variety of conditions, including arthritis, digestive problems, and skin conditions.-*African pepper*: *African pepper*, also known as grains of paradise, is a spice that is used in Benin for its culinary and medicinal properties. It is used to treat a variety of conditions, including digestive problems, fever, and respiratory infections.

The traditional knowledge and uses of these plants have been passed down through generations and are still widely used today in addition to modern medicine.

By studying the biological, toxicological, and phytochemical aspects of medicinal plants, researchers can provide valuable information on the safe and effective use of these plants in traditional medicine. This information can be used to improve health practices, promote the development of the medicinal plant industry, and provide a scientific basis for the recognition and regulation of traditional medicine. Furthermore, this study can also contribute to the preservation of traditional knowledge and the development of the field of ethnobotany, which is the study of the relationships between plants, people, and cultures. By documenting the traditional knowledge of medicinal plants, scientific actors can help ensure that this valuable resource is not lost and can be passed on to future generations. Overall, the study of medicinal plants and their use can have a positive impact on public health, the economy, and the preservation of traditional knowledge, making it an important area of research. According to the World Health Organization (WHO), traditional medicine (TM) is still the first line of health care for more than 80% of people living in developing countries [2]. This medicine, which is based on the endogenous knowledge of medicinal plants, is transmitted from generation to generation [3]. Benin, officially the Republic of Benin, is a West African country bordered by Niger, Burkina Faso, Nigeria, and Togo. Its current population is estimated at 13.71 million, with a GDP growth rate of +5.7% in 2022 [4].

The rich Beninese flora of 2807 plant species offers an interesting ethnobotanical potential and constitutes a valuable source for both practitioners of traditional Beninese medicine and academic researchers [1]. The mission of Adjanohoun et al. [5] listed nearly 501 species that are used as medicinal plants in traditional Beninese medicine.

The WHO recognizes the importance of traditional medicine (TM) as a source of primary health care for many people in developing countries and has called on governments to integrate it into national health systems and policies [6]. To achieve this, the WHO’s strategy for traditional medicine (TM) 2014–2023 was developed in response to a World Health Assembly resolution. The strategic objectives are to help Member States: (i) take advantage of TM’s potential to improve health, well-being, and person-centered health care and (ii) promote the safe and effective use of TM through regulation, research, and integration into health systems [6]. The overall goal is to implement policies and action plans that strengthen the role of TM in maintaining the health of populations. It is suggested that each country establish mechanisms to encourage the use of traditional medicine and make it a complementary tool to modern medicine. This call to action has heightened the focus on medicinal plants of academic researchers and political authorities in Benin. Additionally, Benin has created a favorable political and social environment for the development of traditional medicine and the promotion of medicinal plants.

The Ministry of Health in Benin established a National Program of Pharmacopoeia and Traditional Medicine (PNPMT) which works closely with traditional healers. They were formally organized under the National Association of Traditional Medicine Practitioners of Benin (ANAPRAMETRAB). After a legal battle, they were formally authorized to split into many as associations as they so choose (Law number 1901, regarding the associations). Numerous associations were then created under the banner of two major federations (FaNaMeTraB and FeNaMeTraB). Some traditional healers are not affiliated with any federation; however, they recognize the regulative authority of the Ministry of Health.

Researchers in fields such as ethnopharmacology and pharmacology are actively engaged in various aspects of medicinal plant use. Despite this, there is a significant gap in the commercialization of research results into scientific innovations, such as the development of phytomedicines, with very few standardized herbal medicines being developed from research results to benefit the Beninese population.

This study aims to document the current status of the pharmacological, toxicological, and phytochemical investigations on medicinal plants conducted at the national universities of Benin. Overall, the lack of valorization of research results raises concerns about the quality of research produced in these universities.

## 2. Results

Benin has several universities and higher education institutions that offer programs in various disciplines, including the study of plants:-University of Abomey-Calavi: The University of Abomey-Calavi is the largest and oldest university in Benin, located in the city of Abomey-Calavi. It has several faculties, including the Faculty of Agricultural Sciences, which offers programs in agronomy, plant breeding, soil science, and agroforestry. It has also the Polytechnic School of Abomey-Calavi, which offers programs in a variety of engineering and technology disciplines.-University of Parakou: The University of Parakou is located in the city of Parakou and offers programs in a variety of disciplines, including agriculture and environmental sciences. The Faculty of Agronomic Sciences offers programs in agronomy, agribusiness, and animal production.-National University of Science, Technology, Engineering and Mathematics of Abomey (UNSTIM): The National University of Science, Technology, Engineering and Mathematics of Abomey (UNSTIM) is located in the city of Abomey and offers programs in a variety of disciplines, including agriculture and natural sciences.-National School of Agriculture: The National School of Agriculture is located in the city of Porto-Novo and offers programs in agricultural sciences, including crop production, livestock production, and agribusiness.

In addition to these universities and higher education institutions, Benin also has research centers and institutes that focus on the study of plants, such as the National Institute of Agricultural Research of Benin (INRAB) and the Centre for Agricultural Research in West and Central Africa (WARDA). These institutions conduct research on crops, plant breeding, and plant diseases, and other areas.

In these universities, a total of 274 scientific studies on 269 medicinal plants were identified from 2001 to 2022. Considering the 501 species used in traditional Beninese medicine [5], this total of 269 plants represents 53.69% of these medicinal species.

### 2.1. Biological Activities Studies

A total of 220 scientific studies on the biological activities of 269 medicinal plants were identified. Altogether, these studies cover 29 types of biological activities (Figure 1). The most explored are antimicrobial, antioxidant, insecticidal, antihyperglycemic, and anti-inflammatory activities.

Table 1 highlights the main limitations and areas for improvement in research practices in the field of medicinal plants. Most antimicrobial activities are studied through the in vitro susceptibility testing of microbial strains with disc diffusion methodologies. There is a lack of advanced studies that examine the mechanism of action of the antimicrobial effect and its in vivo efficacy. Additionally, there is sometimes a lack of proper implementation of protocols. Furthermore, the studies conducted thus far only cover microbial growth inhibition, and research on the anti-virulence properties of the medicinal plants has not been carried out.

Regarding antioxidant activity, research on the antioxidant potential of medicinal plant extracts is commonly conducted using in vitro free radical scavenging tests such as the DPPH assay. Limitations in these studies include the lack of multiple tests to confirm the radical scavenging activity, ignorance of other modes of possible antioxidant effect, and a lack of investigation into the in vivo effectiveness of the antioxidant effect.

Few scientific studies have confirmed the anti-inflammatory activity of medicinal plants by researching inflammation mediators. This highlights the need for further research in this area to fully understand the anti-inflammatory effects of these plants.

### 2.2. Toxicological Studies

A total of 107 scientific studies have explored the toxicity of the medicinal plants identified. Most of the studies focused on larval cytotoxicity, followed by acute oral toxicity (Figure 2).

Limitations and opportunities for improving research practices for toxicology studies are presented in Table 2 below. The limitations identified in these studies point to needs for capacity building in advanced toxicity testing of natural substances.

### 2.3. Phytochemical Study

One hundred thirty-eight (138) phytochemical studies were identified among the published reports on medicinal plants. Most of them are related to the sole qualitative profiling by the method of staining and precipitation (Figure 3).

The main limitations of these techniques (Table 3) are related to the frequent use of a preliminary test (Cytotoxicity larceny) without a real deepening. The lack of cytotoxicity assays applied to in vitro cultures of mammalian cell lines limits the ability to predict the toxic potential in the context of human use and necessitates the use of rodents and chicks. This highlights the need to use more advanced methods for the identification and characterization of bioactive molecules.

## 3. Discussion

The aim of this study was to document the current state of scientific research practices regarding the evaluation of the biological, pharmacological, toxicological, and phytochemical properties of medicinal plants in Benin flora. The data obtained indicate that studies conducted in Benin cover priority areas such as pharmacology, toxicology, chemistry, microbiology, etc. In the scientific literature, a study that integrated data from different countries on several other continents documented that these fields constitute the areas in which many studies of medicinal plants are conducted [278]. This finding suggests that the study of medicinal plants in Benin, as well as in other countries outside Africa, primarily focuses on the above-mentioned areas.

The results obtained from this study demonstrate that the scientific studies in Benin concerning these aspects of medicinal plants are still in an embryonic or preliminary stage. In general, there is a lack of scientific research oriented toward more in-depth studies. The current data highlight the character, which is preliminary even at best, of the evidence of confirmation of the therapeutic usefulness of medicinal plants through the study of their biological activity. This situation explains the low availability of phytomedicines resulting from research of a sufficient depth to contribute effectively to the improvement of the health status of the population, more than 80% of whom depend on traditional medicine for their health needs. In this context, scientific elites should fully assume their leadership role by establishing a long-term roadmap to improve the quality of scientific research on medicinal plants and the modes of cooperation between academic staff and traditional medicine practitioners. However, despite their willingness, local research institutions lack adequate technical facilities to work in conditions that allow them to produce quality research results. In Benin, scientific research remains underfunded, with less than 1% of the gross domestic product devoted to it [279].

Furthermore, the results obtained demonstrate that the researchers from the University of Abomey-Calavi were the most active in the scientific investigations mentioned above. This is easily explained by the fact that the University of Abomey-Calavi is the oldest public university in Benin, and its multidisciplinarity is favorable to this state of affairs. At this university, among the biological activities of the medicinal plants explored, antimicrobial activities have been very frequent in both human and animal health. Benin, similar to other developing countries in Africa, has a strong culture in which the population is very attached to the ancestral tradition, which is based on the use of medicinal plants against all kinds of diseases. Thus, among the diseases most targeted by the studies reviewed, infectious diseases ranked in the first position. A similar observation was reported by Salmerón-Manzano et al. [278]. In their study, these authors reported that parasitic diseases were the most frequent in African regions, particularly in Cameroon. These observations can be explained by the fact that infectious diseases constitute one of the deadliest diseases worldwide [280]. In addition, the issue of antimicrobial resistance is a major public health concern at present as it further complicates the management of infectious diseases [281]. The World Health Organization has strongly recommended the exploration of several alternatives to contribute to the effective control of infectious diseases [282], and medicinal plants are among the most-explored alternatives. Therefore, this could justify the commitment of Beninese researchers, particularly those at the University of Abomey-Calavi, to invest in scientific studies examining the antimicrobial potential of medicinal plants.

The impact and interest of a study on the current status of research on medicinal plants at the national universities of Benin is then significant in several ways:-Health improvement: The study can provide valuable insights into the biological, toxicological, and phytochemical aspects of the medicinal plants commonly used in the country, which can inform and improve healthcare practices and policies.-Economic benefits: By identifying the most promising medicinal plants and the most important areas of research, the study can help support the development of the medicinal plant industry in the country, potentially leading to economic benefits.-Scientific advancement: The study can also contribute to the advancement of scientific knowledge in the field of medicinal plants, providing a better understanding of the plants used in traditional medicine and their potential health benefits.-Capacity building: By highlighting the current state of research at national universities, the study can help identify areas where capacity building is needed and inform future investment in research and development.

Overall, the study can have a positive impact on public health, the economy, and scientific research in Benin, and it could be of interest to researchers, healthcare providers, policymakers, and other stakeholders in the region.

## 4. Materials and Methods

The data collection and analysis were conducted over a 4-month period (September to December 2022) via a literature review. The process was organized in three main steps:-The first step was to search the following search engines (Google Scholar, Pubmed, Sciencedirect, and FreeFullPDF) for scientific information on the medicinal plants studied in Benin, using the following expressions or word groups: “Biological activities”; “Pharmacological activities”; “Toxicity”; “Phytochemistry” “Phytochemical screening”, and “Quantitative screening”. This search was performed in both French and English. The information was collected according to each of the four major universities of Benin (the University of Abomey-Calavi, University of Parakou, National University of Agriculture of Kétou, and the National University of Science, Technology, Engineering and Mathematics of Abomey).-The second step consisted of a critical analysis of the methodological approach of each of the scientific studies identified in order to identify the main limitations of the methodologies used and to propose possible actions to improve research practices. For the scientific articles whose authors were in collaboration via several research institutions, the authorship of the article was defined by considering the corresponding author from one of the national universities of Benin.-The last step consisted of making a synthesis of the information collected about the different pharmacological, toxicological, and phytochemical activities by university.

The Table 4 presents the description of the articles included in the study.

## 5. Conclusions

Faced with this alarming situation, it is urgent that Beninese researchers implement mechanisms for the real development of science in Benin. Opportunities for North–South and other partnerships are desired more than ever to strengthen the technical capacities of research structures and to strengthen the skills of teacher-researchers on the methodological approaches currently accepted in the biomedical field, especially in those of the development of plant-based remedies and standardized phytomedicines.

## Figures and Tables

**Figure 1 plants-12-01506-f001:**
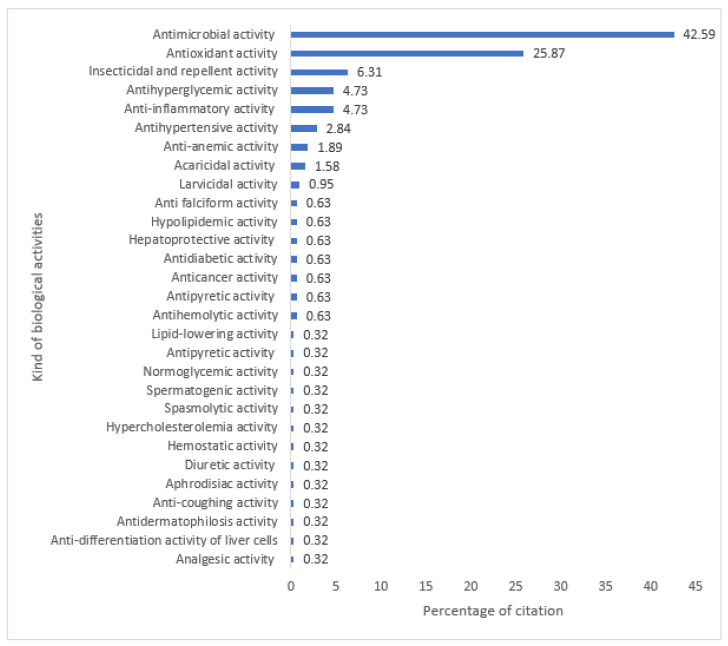
Summary of pharmacological studies carried out on different medicinal plants.

**Figure 2 plants-12-01506-f002:**
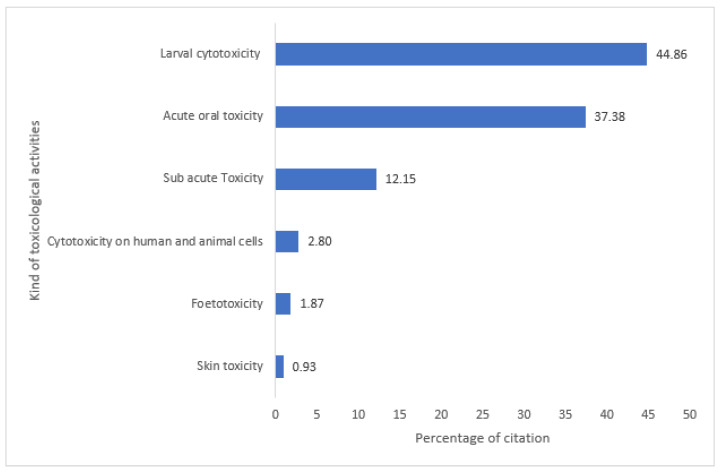
Summary of toxicology studies carried out on medicinal plants.

**Figure 3 plants-12-01506-f003:**
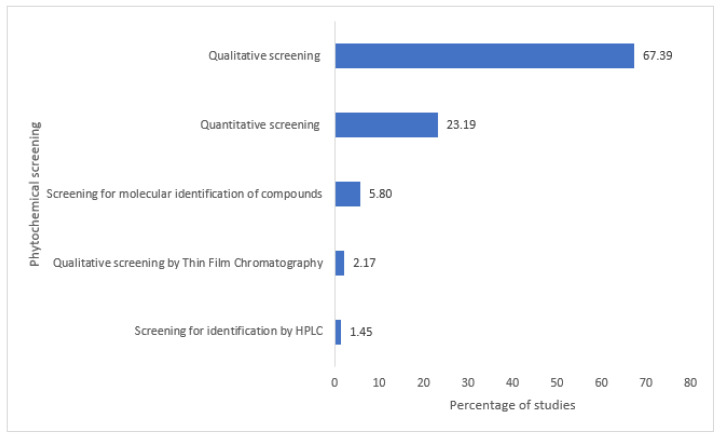
Summary of phytochemical studies of medicinal plants.

**Table 1 plants-12-01506-t001:** Limitations and suggestions for improving research practices in relation to the biological activities explored.

Biological Activities	Methodology	Limitations–Prospects	Improvement of Research Practice	References
Antimicrobial activity	- In vitro testing of the sensitivity of microbial strains to plant extracts using diffusion or micro-dilution; - Larval migration inhibition test (LMIT): incubating larvae with the extracts at different concentrations; - Incorporation of dried seed powder in chicken feed, followed by fecal analysis; - Contact mortality test by exposing worms collected from the abomasum of naturally infested sheep to six (6) different concentrations (4, 8, 16, 32, 64, and 128 mg/ml) of the plant extract and to a reference molecule, levamisole; Artificial infestation of day-old chicks with spore-forming oocysts of Eimeria tenella, followed by treatment with a decoction of the plant and subsequent analysis of feces to assess reduction in oocyst excretion. Monitoring of the parasite load in lambs naturally and artificially infested with *Haemonchus contortus* and *Trichostrongylus colubriformis* larvae, then treated with dry leaf powders on one hand and consumption of fresh leaves of the plant on the other.	Research of the mechanism of the antibacterial effect, of the in vivo efficiency; - Lack of precision on the concentration and dose of the reference molecule used; - Mechanism of action not elucidated; - Risk of bias, due to the fact that the worms tested did not have the same vitality, especially as some are placed in PBS to be revitalized, and mechanism of action not established; - The method does not ensure uniformity of dose or amount of feed ingested to make optimal comparisons at batch level; - No explanation for the choice of the chosen doses.	Training on advanced tests to elucidate the mechanism of action of the antibacterial effect (in vitro and in vivo). Capacity building on standardised methods for elucidating the mechanism of pest control action. Capacity building on standardised methods for conducting in vivo anti-parasitic activity. Capacity building on the number of tests required to validate the anti-nematode activity of medicinal plant extracts. Capacity building on pharmacological evaluation methods. Dose selection and objective assessment of results.	[7,8,9,10,11,12,13,14,15,16,17,18,19,20,21,22,23,24,25,26,27,28,29,30,31,32,33,34,35,36,37,38,39,40,41,42,43,44,45,46,47,48,49,50,51,52,53,54,55,56,57,58,59,60,61,62,63,64,65,66,67,68,69,70,71,72,73,74,75,76,77,78,79,80,81,82,83,84,85,86,87,88,89,90,91,92,93,94,95,96,97,98,99,100,101,102,103,104,105,106,107,108,109,110,111,112,113,114,115,116,117,118,119,120,121,122,123,124]
Acaricidal activity	In vitro test on the survival of ticks exposed to various concentrations of an extract.			[125,126]
Antioxidant activity	In vitro test, mostly using the DPPH method.	Research on the mechanism of the antioxidant effect, in vivo antioxidant effect. Failure to respect the number of tests required to validate the antioxidant activity.	Capacity building on the number of tests required to validate the antioxidant activity of medicinal plant extracts, their implementation, and the procedures for the in vivo exploration of the antioxidant effect with the key to the mode of action of this effect.	[11,13,14,15,19,20,34,35,37,39,43,53,64,70,71,73,74,88,88,91,103,105,106,110,127,128,129,130,131,132,133,134,135,136,137,138,139,140,141,142,143,144,145,146,147,148,149,150,151,152,153,154,155,156,156,157,158,159,160]
Anti-inflammatory activity	In vitro tests for the inhibition of enzymatic activity are the most widely used. In vivo test on rats by the administration of an extract, followed by the determination of the white blood cell count.	Research of the in vivo anti-inflammatory effect. No assay of inflammation markers.	Capacity building on the research of the in vivo mechanism of action of the anti-inflammatory effect of extracts. Capacity building on in vivo testing for inflammation markers.	[15,106,138,149,161,162,163,164,165,166]
Insecticidal and repellent activity	Multi-dose immersion test with increasing progression. Application of leaf powder directly on the larvae and spraying solutions with a hand sprayer in the boxes (in the laboratory).	Lack of justification for the infesting dose. No precision on the volume of extract sprayed.	Capacity building on the conduct of field and laboratory testing of insecticidal activity.	[167,168,169,170]
	Spraying extracts on plant organs (field experimentation).	No details on the volume of extract sprayed.	Capacity building on the conduct of field and laboratory testing of insecticidal activity.	[170,171]
Antihypertensive activity	In vivo test on Wistar rats with the exploration of few relevant parameters.	Advanced study with the consideration of several parameters to elucidate the mechanism of action.	Capacity building on adapted models to elucidate the mechanism of action.	[172,173,174,175,176,177,178,179]
Antihemolytic activities	In vitro assay on erythrocytes	.		[180,181]
Antihyperglycemic activity	In vivo test on Wistar rats, followed by blood glucose determination. In-vivo testing on rats by the induction of metabolic disorder and the administration of extract concentrations and an evaluation of the effects.	Advanced study via more adapted models to elucidate the anti-diabetic effect of medicinal plants. Use of old model methods at the expense of more advanced methods.	Capacity building on advanced studies using more suitable models to elucidate the anti-diabetic effect of medicinal plants Capacity building on the design and execution of modern research protocols in pharmacology.	[146,182,183,184,185,186,187,188,189,190,191,192,193,194]
Analgesic activity	In vivo test on Wistar rats using the tail immersion method.	Advanced study via more adapted models.	Training on advanced tests to explore the analgesic effect.	[162]
Anti-differentiation or anticancer activity of liver cells	In vivo test on Wistar rats with methodological deficiencies.	Advanced study via more adapted models.	Training on advanced tests to explore the anticancer effect of medicinal plants.	[195,196,197,198]
Anti-anemic activity	- -Osmotic resistance test of erythrocytes from Wistar rats. - -In vivo test on Wistar rats.	Lack of precision on the justification of the doses used.	Capacity building on the design and execution of pharmacology research protocol.	[199,200,201,202,203]
Anti-dermatophilosis activity	Skin test on cattle with dermatophilosis and scabby lesions.			[204]
Anti-diabetic activity	Diabetes induction model with streptozotocin.	Advanced study on the research of the mechanism of action of the antidiabetic effect.	Capacity building on advanced studies to investigate the mechanism of action of the anti-diabetic effect.	[110,161,205,206,207,208]
	In vivo test on normal rats without diabetes induction.	The experimental design is not consistent with the anti-diabetic effect being explored.	Capacity building on appropriate tests for diabetes induction and the exploration of the antidiabetic effects of plant extracts.	[209,210]
Insecticidal and repellent activity	Test on the pest *D. porcellus*.			[211]
Larvicidal activity	In vitro test on larvae.			[212,213,214]
Anti-coughing activity	In vivo test on guinea pigs using the citric acid method.			[215]
Aphrodisiac activity	In vivo test with exploration of the parameters related to the aphrodisiac effect.			[216,217,218]
Diuretic activity	- In vivo test with little data on some important aspects of pharmacology. - No exploration of enzymatic parameters.	Advanced study on the most suitable models.	Training on advanced study via more suitable models.	[219,220]
Hemostatic activity	In vitro (anti-coagulation) and in vivo test to stop bleeding.			[221,222,223,224]
Hepatoprotective activity	In vivo test on Wistar rats after the induction of liver damage, followed by an evaluation of enzymatic and histological parameters.	Advanced study on adapted models allowing the elucidation of the mechanism of action. No assay of inflammation markers.	Training on advanced study via more suitable models. Capacity building on in-vivo tests for inflammation markers.	[194,225,226]
Hypercholesterolemia activity	Induction of hypercholesterolemia, followed by measurement of blood parameters.	Advanced study on the elucidation of the mechanism of action.		[227]
Spasmolytic activity	Tested on isolated rat trachea in combination with an aqueous extract of *Afromomum melegueta.*			[228]
Spermatogenic activity	In vivo test on Wistar Rats, followed by assessment of reproductive parameters.	Advanced study on other models allowing the elucidation of the mechanism of action.	Training on advanced study via other models to elucidate the mechanism of action.	[229,230]
Normoglycemic activities	In vivo test on Wistar rats, followed by blood glucose determination.	Advanced study via more adapted models to elucidate the anti-diabetic effect of medicinal plants.	Capacity building on advanced studies using more suitable models to elucidate the anti-diabetic effect of medicinal plants.	[182]
Lipid-lowering activity	In-vivo test on rats by forced administration of 2 ml of pork fat and sugar water.	More suitable models should be considered.	Capacity building on the design and execution of the research protocol in pharmacology.	[231]
Antipyretic activity	In vivo test on Wistar Rats.	Advanced study on antipyretic properties.	Advanced study training via other models.	[162,200]
Antidiarrheal activity	In vivo test on Wistar Rats.	Advanced study on antidiarheal properties.	Advanced study training via other models.	[232]
Anticonvulsiant activity	In vivo test.			[152]

**Table 2 plants-12-01506-t002:** Methodologies of studies of toxicity tests: limitations and proposals for improvement of research practice.

Toxicological Study	Methodology	Limitations—Prospects	Improvement of Research Practices	References
Larval cytotoxicity	In vitro test on the survival of *Artemia salina* larvae exposed to various concentrations of the extract.	Subjective assessment of the results. In vivo toxicological, cellular, and genotoxicity studies; research on heavy metals.	Capacity building on advanced in vivo, cellular and genotoxicity toxicology tests; heavy metals research.	[3,19,23,28,29,39,42,77,78,91,96,98,129,132,138,140,142,174,190,219,233,234,235,236,237,238,239,240,241,242]
Acute oral toxicity	In vivo single-dose test, followed by an exploration of clinical signs of toxicity after 14 days in accordance with OECD 413.	Advanced toxicity tests: cellular and genotoxicity; heavy metals research. The standard used does not correspond to the acute oral toxicity test.	Capacity building on advanced in vivo, cellular and genotoxicity toxicology tests; heavy metals research Training on the implementation of protocol 423 which is adapted to the acute oral toxicity test.	[3,44,53,63,113,146,182,189,200,236,240,243,244,245,246,247,248,249,250,251,252]
Cellular toxicity test	HepG2 cell test.	No limit noted.		[159]
Acute toxicity test with 423 OECD	*In vivo* single-dose test followed by investigation of clinical signs of toxicity after 14 days in accordance with OECD 423.	No limit noted.		[191,192,209,253,254,255,256,257,258,259]
Foetotoxicity	Test investigating the effect of plant extracts on the embryonic development of chicks.	Advanced toxicity tests: cellular and genotoxicity; heavy metals research.	Capacity building on *advanced* in vivo, cellular and genotoxicity toxicology tests; heavy metals research.	[260,261]
Sub-acute toxicity	In vivo test on rats, not according to the 407 OECD.	Non-compliance with the required standard in relation to chronic toxicity. Training on toxicity tests according to the required standards.	Capacity building on advanced in vivo, cellular and genotoxicity toxicology tests; heavy metals research.	[191,192,235,253,254,258,259,262,263,264,265,266]
Skin toxicity	Skin testing on Wistar rats.	Advanced toxicity tests: cellular and genotoxicity; heavy metals research.	Capacity building on advanced in vivo, cellular and genotoxicity toxicology tests; heavy metals research.	[267]
Cytotoxicity on human and animal cells	Test on human and animals cells.			[268]
	Cytotoxicity in WI38 non-cancerous fibroblast cells using the MTT assay.	No limit.	Training on standards-based toxicity testing.	[118]

**Table 3 plants-12-01506-t003:** Limitations of the phytochemical study methodology and proposals for practice improvement.

Phytochemical Study	Methodology	Limitations-Prospects	Improvement of Research Practices	References
Qualitative screening by staining and precipitation	Detection of the presence of major phytochemical groups, using the precipitation and staining method.	Subjective assessment of the results. Perspective of phytochemical study aiming at the identification and characterization of bioactive compounds of plant extracts.	Training on the use of chromatographic and spectroscopic methods for the characterization of bioactive compounds in plant extracts.	[13,19,20,23,28,29,38,43,44,53,61,67,70,72,74,76,77,78,82,89,90,91,92,93,103,104,105,106,107,108,109,110,111,120,125,132,133,141,142,145,146,148,149,150,151,152,153,157,160,163,172,176,182,187,188,190,192,193,200,220,221,233,234,239,241,243,252,266,269,270,271,272,273,274]
Screening for molecular identification of compounds	Identification of compounds by GC and GC/MS.			[73,96,129,156,163,268,275,276]
Qualitative screening by thin-layer chromatography	Detection of the presence of major phytochemical groups.	Lack of precision in the recognition of certain phytochemical groups. Prospects for phytochemical studies aimed at the identification and characterization of bioactive compounds in plant extracts.	Training on the use of chromatographic and spectroscopic methods for the characterization of bioactive compounds.	[71,156,173,270]
Screening for HPLC identification	Identification of molecules using HPLC.			[40,148,155]
Quantitative screening of polyphenols or other secondary metabolites	Spectrophotometric method.	Perspective of phytochemical study aiming at the identification and characterization of bioactive compounds of plant extracts.	Training on the use of chromatographic and spectroscopic methods for the characterization of bioactive compounds in plant extracts.	[42,78,92,93,103,105,106,108,109,118,132,137,144,145,146,148,151,154,158,277]

**Table 4 plants-12-01506-t004:** Update on the scientific articles included in this study.

Total Number of Articles Included in This Study	272	
Year range	2001–2022	
Total number of plants	269	
Total number of articles per University	University of Abomey-Calavi	218
	National University of Science, Technology, Engineering and Mathematics of Abomey	35
	University of Parakou	7
	National University of Agriculture	12
Number of publications by language	English	251
	French	21

## Data Availability

All data generated and/or analyzed during the current study are included in this published article. The datasets used and/or analyzed during this study are also available from the corresponding author upon reasonable request.

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
