# Peer review of "Unleashing the Potential of Medicinal Plants in Benin: Assessing the Status of Research and the Need for Enhanced Practices"

_plants, 2023, doi:10.3390/plants12071506_

Round 1
Reviewer 1 Report
Comments:
It is an excellent work that does not need modification, just a suggestion. Place a section where you indicate the aspect of infrastructure and where you get your resources for the development of research.
Author Response
Answers
Thank you very much for the great comment that shows the importance of this work for the community of researchers. Regarding the suggestion, it has been included in the funding section (Page 28). The research design also has been improved in the text. Thank you again.

Reviewer 2 Report
The manuscript with the title "Report on the Current Status of Research on Biological, Toxicological, and Phytochemical Aspects of Medicinal Plants in Benin's National Universities" is an important part of a program that aims to inventory the chemical, biological, toxicological studies of plant resources in Benin with their shortcomings. This work is very important and complex, and the approach to the theme seems very interesting and with new elements. I would have a few small observations:
- Latin words - should be written in italics (e.g. in vivo, etc.)
In the Introduction - to add a small paragraph with the most used plants by the population of Benin and their uses: empirical, scientific, etc.
- I would add at the beginning of the RESULTS chapter, some information about the universities in Benin (their number, localities, faculty or disciplines that dealt with the study of plants)
- In the name of each table, enter the name of the university In each table, please also enter the studied medicinal plants (in particular, those on which several studies have been done).
- In Table 13- which represents the conclusions of these studies, besides no. total number of plants, etc., please also enter the no. universities studied and no. article/university (top of universities).
Author Response
Answers
Thank you very much for the great comment that shows the importance of this work for the community of researchers.
Your review was very positive and we are happy to address the comments :
- Latin words - should be written in italics (e.g. in vivo, etc.)
This has been done through the whole text.
In the Introduction - to add a small paragraph with the most used plants by the population of Benin and their uses: empirical, scientific, etc.
We inserted it at the page 2. The text inserted is as follow :
Benin is a country in West Africa with a rich tradition of herbal medicine. Many plants are used by the population of Benin for various purposes, including medicinal, culinary, and spiritual uses. Here are some of the most commonly used plants in Benin:
Aloe vera: Aloe vera is a succulent plant that is commonly used for medicinal purposes. In Benin, it is used to treat various ailments, including skin conditions, digestive problems, and infections.
Neem: Neem is a tree that is widely used in Benin for its medicinal properties. It is used to treat a variety of conditions, including fever, malaria, and skin diseases.
Hibiscus: Hibiscus is a plant that is used in Benin to make a popular drink known as bissap. The drink is made by steeping the leaves of the hibiscus plant in hot water and adding sugar or honey. It is believed to have a range of health benefits, including reducing high blood pressure and boosting the immune system.
Moringa: Moringa is a tree that is used in Benin for its nutritional and medicinal properties. It is rich in vitamins and minerals and is used to treat a variety of conditions, including anemia, diabetes, and high blood pressure.
Baobab: Baobab is a tree that is native to Africa and is used in Benin for its medicinal and nutritional properties. The fruit of the baobab tree is rich in vitamin C, fiber, and antioxidants, and is used to treat a variety of conditions, including diarrhea, constipation, and respiratory infections.
Ginger: Ginger is a root that is widely used in Benin for its medicinal and culinary properties. It is used to treat a variety of conditions, including nausea, indigestion, and inflammation.
Turmeric: Turmeric is a spice that is widely used in Benin for its medicinal and culinary properties. It is used to treat a variety of conditions, including arthritis, digestive problems, and skin conditions.
African pepper: African pepper, also known as grains of paradise, is a spice that is used in Benin for its culinary and medicinal properties. It is used to treat a variety of conditions, including digestive problems, fever, and respiratory infections.
These are just a few examples of the plants commonly used in Benin. The traditional knowledge and uses of these plants have been passed down through generations and are still widely used today, alongside modern medicine.
- I would add at the beginning of the RESULTS chapter, some information about the universities in Benin (their number, localities, faculty or disciplines that dealt with the study of plants)
This comment has been inserted. The text is as followed :
Benin has several universities and higher education institutions that offer programs in various disciplines, including the study of plants. Here is some information about the universities in Benin:
University of Abomey-Calavi: The University of Abomey-Calavi is the largest and oldest university in Benin, located in the city of Abomey-Calavi. It has several faculties, including the Faculty of Agricultural Sciences, which offers programs in agronomy, plant breeding, soil science, and agroforestry.
University of Parakou: The University of Parakou is located in the city of Parakou and offers programs in a variety of disciplines, including agriculture and environmental sciences. The Faculty of Agronomic Sciences offers programs in agronomy, agribusiness, and animal production.
University of Abomey: The University of Abomey is located in the city of Abomey and offers programs in a variety of disciplines, including agriculture and natural sciences.
National School of Agriculture: The National School of Agriculture is located in the city of Porto-Novo and offers programs in agricultural sciences, including crop production, livestock production, and agribusiness.
In addition to these universities and higher education institutions, Benin also has research centers and institutes that focus on the study of plants, such as the National Institute of Agricultural Research of Benin (INRAB) and the Centre for Agricultural Research in West and Central Africa (WARDA). These institutions conduct research on crops, plant breeding, and plant diseases, among other areas.
- In the name of each table, enter the name of the university In each table, please also enter the studied medicinal plants (in particular, those on which several studies have been done).
The comments have been inserted fully for the first part. However, this will be very huge to present those plants in the tables. People will get confused. That is why we submitted the full database as supplementary file.
- In Table 13- which represents the conclusions of these studies, besides no. total number of plants, etc., please also enter the no. universities studied and no. article/university (top of universities).
The comment has been addressed and inserted (Page 4)

Reviewer 3 Report
Review of
Report on the Current Status of Research on Biological, Toxicological, and Phytochemical Aspects of Medicinal Plants in Benin's National Universities
This is an interesting paper, the study is relevant, and the authors contribute to this field of research.
A few corrections are necessary:
Results: In relation to figures 2, 3, 4, 5, 6, 7, 8,9 the graphical representation mode is not relevant, the values are not represented. Another way of representation must be chosen
Discussions: the results obtained must be correlated with the results obtained in similar studies
References: The authors need to check the list of references and correct it according to the Instructions for authors.
Author Response
Answers
Results: In relation to figures 2, 3, 4, 5, 6, 7, 8,9 the graphical representation mode is not relevant, the values are not represented. Another way of representation must be chosen
All the figures have been changed.
Discussions: the results obtained must be correlated with the results obtained in similar studies
The discussion has been improved.
References: The authors need to check the list of references and correct it according to the Instructions for authors.
The references have been extensively updated.

Round 2
Reviewer 3 Report
Dear Authors,
The previously mentioned figures have been modified.
Please also present the obtained values on these figures
Figures 6, 8 and 9 show only two values, which are mentioned in the text, so they are not relevant for the presented results. My opinion is that they can be deleted.
Author Response
We have deleted figures 6,8 and 9 as suggested by the reviewer. We also present the values on the figures.
